# Observation and rationalization of nitrogen oxidation enabled only by coupled plasma and catalyst

Hanyu Ma [1], Rakesh K. Sharma[2], Stefan Welzel[2], Mauritius C. M. van de Sanden [2,3], Mihalis N. Tsampas [2✉] & William F. Schneider [1,4✉]

Heterogeneous catalysts coupled with non-thermal plasmas (NTP) are known to achieve reaction yields that exceed the contributions of the individual components. Rationalization of the enhancing potential of catalysts, however, remains challenging because the background contributions from NTP or catalysts are often non-negligible. Here, we first demonstrate platinum (Pt)-catalyzed nitrogen ($N_2$) oxidation in a radio frequency plasma afterglow at conditions at which neither catalyst nor plasma alone produces significant concentrations of nitric oxide (NO). We then develop reactor models based on reduced NTP- and surface-microkinetic mechanisms to identify the features of each that lead to the synergy between NTP and Pt. At experimental conditions, NTP and thermal catalytic NO production are suppressed by radical reactions and high $N_2$ dissociation barrier, respectively. Pt catalyzes NTP-generated radicals and vibrationally excited molecules to produce NO. The model construction further illustrates that the optimization of productivity and energy efficiency involves tuning of plasma species, catalysts properties, and the reactor configurations to couple plasma and catalysts. These results provide unambiguous evidence of synergism between plasma and catalyst, the origins of that synergy for $N_2$ oxidation, and a modeling approach to guide material selection and system optimization.

[1] Department of Chemical and Biomolecular Engineering, University of Notre Dame, Notre Dame, IN 46556, USA. [2] Dutch Institute for Fundamental Energy Research (DIFFER), De Zaale 20, 5612 AJ Eindhoven, The Netherlands. [3] Department of Applied Physics, Eindhoven University of Technology (TU/e), 5600 MB Eindhoven, The Netherlands. [4] Department of Chemistry and Biochemistry, University of Notre Dame, Notre Dame, IN 46556, USA. ✉email: m.tsampas@differ.nl; wschneider@nd.edu

The ability of a nonthermal plasma (NTP) and heterogeneous catalyst combination to achieve reaction yields that exceed the contributions of the individual components is well documented[1,2]. Often reactions are explored at conditions at which NTP or thermal catalytic yields are non-negligible. Disentangling the gain achieved by combining NTP and catalyst from the background contributions of NTP and catalyst alone, and inferring the origins of yield enhancements, are thus significant practical challenges[3–5]. Here, we demonstrate NTP-catalytic nitrogen oxidation:

$$N_2(g) + O_2(g) \leftrightarrow 2NO(g) \qquad (1)$$

in a reactor configuration and at conditions in which neither catalyst nor NTP yields a significant product. Observed NO production is thus the result of the mutual action of NTP and catalyst. Further, we demonstrate a modeling strategy to integrate and isolate NTP and catalyst contributions to observed performance. These models recover and rationalize the observed productivity and provide a foundation for system optimization.

Reaction (1) is highly endothermic, and at ambient conditions in air, the equilibrium lies far to the left (Supplementary Fig. 1). The equilibrium shifts towards NO with increasing temperature, a fact exploited in the Birkeland-Eyde (B-E) process for thermally fixing $N_2$ at high temperatures achieved within a thermal plasma[6]. At these high temperatures, $N_2$ and $O_2$ are partially atomized, and NO is produced by O and N radical reactions with $N_2$ and $O_2$, respectively, in the so-called Zeldovich mechanism[7]. Similar processes are at play in the conventional combustion of fuel in the air, motivating interest in NO decomposition catalysts for environmental protection[8]. However, catalytic $N_2$ oxidation under thermal conditions is unknown.

Nonthermal $N_2/O_2$ plasmas are known and have been observed to generate NO at bulk gas temperatures much below those necessary for thermal $N_2$ oxidation. NO concentrations can even exceed those expected based on the bulk thermodynamic equilibrium[9]. These NTPs contain ground and excited neutrals, radicals, ions, and electrons, and kinetic models that incorporate the reactions of these active species have been applied to microwave[10,11], pulsed-power gliding-arc[12], glow discharge[13], and stationary plasmas[14]. These models recover observed NO concentrations and even the densities of intermediates[10,12,13]. These models suggest that the same Zeldovich mechanism is at play in NTPs as in thermal $N_2$ oxidation[15,16].

NO yields have been reported to increase over a non-zero background when a radio frequency, microwave or dielectric barrier discharge plasma is combined with a catalyst[17–19]. Patil et al. showed that the extent of that increase is catalyst-dependent, in experiments comparing a variety of metal oxides within a packed-bed, dielectric barrier discharge reactor[19]. Similarly, the introduction of $WO_3$ and $MoO_3$ downstream of either an inductively coupled high-frequency plasma or microwave plasma was observed to increase NO yields over the background[18,20]. In all cases, the observed improvements are with respect to significant plasma-only NO yields. Models to qualitatively or quantitatively explain these observations have not been reported.

These observations leave open the question of the mechanisms by which NO production is increased and even the extent to which that increase can be attributed to surface catalytic reactions. To disentangle the NTP and catalytic contributions to $N_2$ oxidation, here we report $N_2$ oxidation experiments at low $O_2/N_2$ mixing ratios at which NTP-only and Pt-catalyzed NO productivity are each vanishingly small. We demonstrate that the introduction of a Pt catalyst positioned postdischarge from the $N_2$-$O_2$ radio frequency plasma results in a substantial increase in NO production. These results provide unambiguous evidence that productivity arises solely from the synergy between plasma and catalyst, an observation unique to the literature. They further enable the development of microkinetic models that quantitatively rationalize the catalytic origins of productivity and its sensitivity to operating conditions.

To interpret and quantify this evident synergy between NTP and catalyst, we represent the experimental system as a series of coupled reactors to predict NO production in the absence and presence of a Pt catalyst. Instead of explicitly estimating the plasma characteristics with detailed plasma models, we describe the plasma chemistry using a combination of vibrational heating of the diatomics and atomic nitrogen generation and parameterize surface reactions with density-functional-theory (DFT)-computed data. The reduced models capture the essentials of the product formation behaviors and enable the identification of regions in which coupled plasma and catalyst enhances or has a negligible to deleterious effect on NO generation. At $O_2$ mole fractions on the order of $10^{-3}$, plasma-only NO generation is suppressed by the reverse Zeldovich reactions, while thermal catalytic NO production is prevented by the inability of Pt to dissociatively adsorb $N_2$. In contrast, Pt can catalyze the reaction of NTP-generated N radicals and vibrationally excited $N_2$ with surface-bound O to generate NO. This synergistic effect is most pronounced at conditions at which neither individual component is effective and diminishes at conditions at which oxygen dominates the surface chemistry, consistent with experimental observation here and elsewhere[17–19]. The results highlight approaches to exploiting plasma-catalytic chemical synthesis as well as modeling strategies for identifying optimal plasma-catalyst combinations.

## Results and discussion

**Plasma-catalytic $N_2$ oxidation experiments**. We measured NO production via $N_2$ oxidation in a radio frequency plasma reactor at low $O_2$-to-$N_2$ pressure ratios, with and without downstream catalyst. The reactor consists of an inductive coil connecting to a radio frequency power supply and a matching network and a quartz tube with a heating mantle (Fig. 1a). A porous Pt film deposited on a tubular YSZ membrane is used as the catalyst. Catalyst microstructure consists of a network of percolated particles of the order of micron (Fig. 1b) with the thickness of approximately 14 micron (Fig. 1c). $N_2$-$O_2$ mixtures are introduced to the reactor at 100 SCCM, 5 mbar, and ambient temperature.

The $N_2$-$O_2$ plasma is generated in the area near the coil and activated species flow toward the heating mantle, which is kept at 873 K[21]. Figure 1d reports observed NO concentrations in the absence and presence of catalyst as a function of inlet $O_2$ mole fractions from $10^{-4}$ to $10^{-2}$. Also reported is the thermodynamic equilibrium production of NO at 873 K. NO production at zero plasma power (i.e. thermal catalysis) is zero within the detection limit of the instrument. Thus thermal catalysis is ineffective at these operating conditions. Moreover, NO production via plasma (80 W) is also ineffective. NO concentrations are less than 20 and 60 ppm without and with the YSZ tube with a standard deviation of 40–50%.

However, NO production increases significantly upon placing a Pt catalyst in the middle of the heating mantle (Fig. 1d). NO concentrations exceed thermal equilibrium across the entire gas composition range explored and vary nonlinearly with $O_2$ mole fraction, maximizing near $5 \times 10^{-4}$ and decreasing at lower or higher $O_2$ concentrations. The results provide unambiguous evidence of NO production dependent upon both plasma and Pt catalyst and sensitivity to exact plasma composition. X-ray diffraction (Supplementary Fig. 3a) and X-ray photoelectron spectroscopy (Supplementary Fig. 3b) observations confirm that

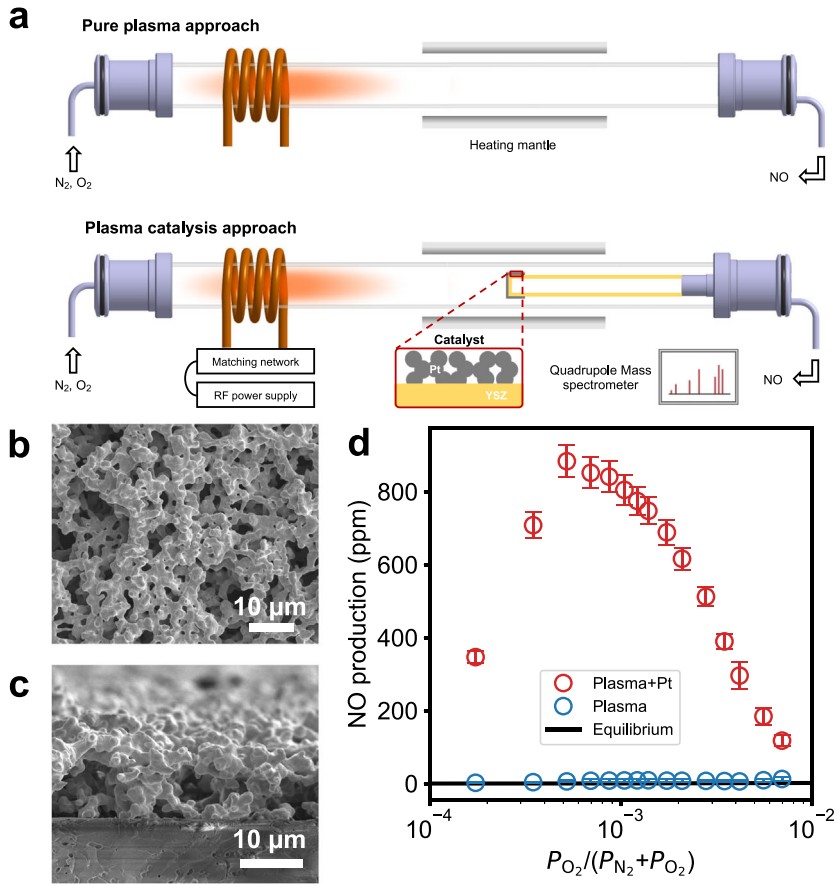

**Fig. 1 Plasma-catalytic N$_2$ oxidation experiments. a** Schematic representation of the plasma reactor setup in a plasma-only configuration (top) and plasma catalytic mode (bottom). In the catalytic mode, a porous Pt film is coated on the end of a non-porous, capped yttria-stabilized zirconia (YSZ) tube. SEM images of (**b**) surface and (**c**) cross-sectional micrographs of porous Pt film deposited on YSZ substrate. **d** Steady-state NO production from N$_2$ and O$_2$ radio frequency postdischarge at 873 K with low O$_2$ pressures. The error bars represent the standard deviation of at least three measurements.

the bulk and surface of the Pt catalyst are unmodified at all tested O$_2$ mole fractions in Fig. 1 by a plasma exposure of 30 min for each measurement.

**Plasma-catalytic N$_2$ oxidation models**. To rationalize plasma-catalytic NO production and its unusual dependence on O$_2$ mole fraction, we created microkinetic models for the thermal catalytic, non-thermal plasma, and coupled systems. Figure 2 illustrates the relevant physical processes in each case, including surface activation of thermalized gas molecules, reactions of vibrationally excited molecules and of radicals present in an NTP, and reactions at the interface between the two, respectively. Catalytic reactions occur at surfaces and thus are characterized by rates per surface site, or turnover frequency. Plasma-phase reactions occur in an (inhomogeneous) bulk phase. The relative contributions of the two to observed product concentrations are thus dependent on the relative number of active site and volume of the reactor. Here, we first consider intrinsic rates over a catalyst in the absence and presence of relevant concentrations of plasma-generated, excited species. We then couple the two through an integral reactor series parameterized to be representative of the reactor of Fig. 1a and incorporating plasma-only and plasma-catalytic steps.

*Intrinsic catalytic rates*. Catalytic N$_2$ oxidation is the reverse of the more widely studied catalytic NO decomposition reaction. The overall reaction energy and free energy are both about 1.8 eV because the reaction conserves molecules and therefore $\Delta S° \approx 0$.

Thermal catalytic N$_2$ oxidation is therefore endergonic. We adopt as the thermal catalytic mechanism reactions indicated to be relevant to NO decomposition on Pt[22–24]. Figure 3a summarizes the potential energy surfaces for N$_2$ oxidation over models for a Pt terrace (Pt(111)) and a step (Pt(211)), extracted from previously reported DFT results[24]. The initial N$_2$ activation step is both endothermic and has high barrier on terraces and even step sites on Pt. We supplement this reaction scheme with two additional steps to incorporate the potential adsorption of plasma-generated radicals, consistent with their observed relevance to plasma-wall chemistry[15,25]:

$$N(g) + * \leftrightarrow N* \qquad (2a)$$

$$O(g) + * \leftrightarrow O* \qquad (2b)$$

where * represents a surface active site. Kinetic parameters for all surface reaction steps are detailed in Supplementary Table 1.

We parametrize a mean-field microkinetic model to predict the intrinsic steady-state NO turnover frequency (TOF) over Pt at conditions consistent with experiment (Fig. 1). TOFs are computed at fixed N$_2$, O$_2$, N, and O pressures and at zero conversion[26,27]. N radical densities are estimated from N$_2$ dissociation fractions measured at similar plasma conditions and O$_2$ dissociation fractions assumed to be one order of magnitude greater than N$_2$[28–32]. N$_2$ and O$_2$ are assumed to have the same vibrational temperature, and the vibrational energy distributions and the consequent effects on the kinetics of N$_2$ and

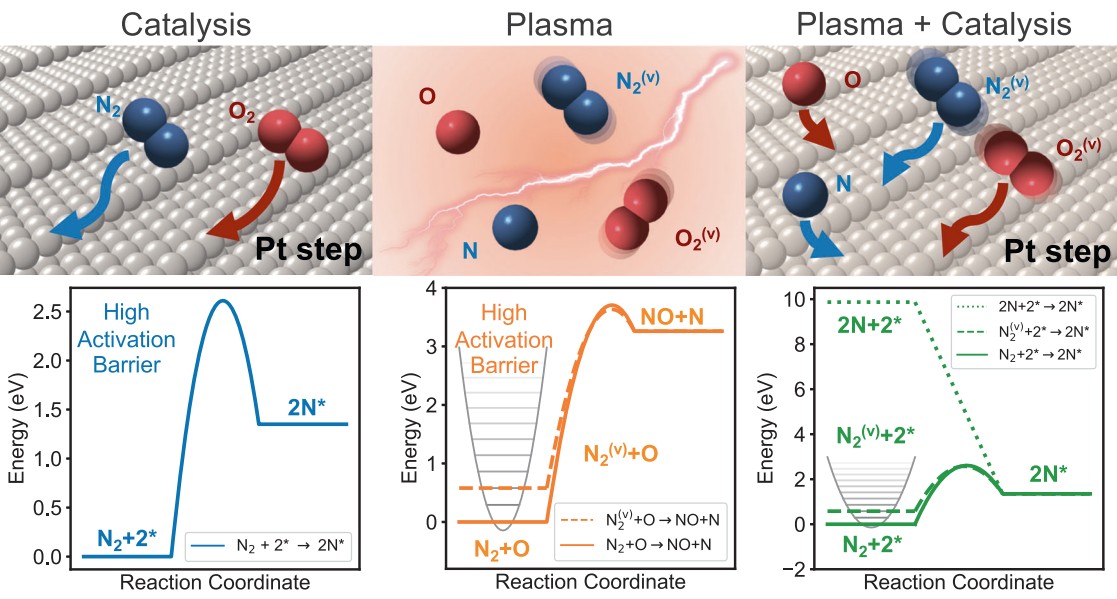

**Fig. 2 N₂ activation pathways.** Schematic representations of active species (top) and potential energy surfaces of different $N_2$ activation processes (bottom) in thermal catalysis, non-thermal plasma and non-thermal plasma coupled with catalysis.

$O_2$ dissociative adsorption are modeled following Mehta et al. (Supplementary Methods and Supplementary Fig. 4)[33].

Figure 3b compares TOFs at a Pt step as a function of conditions. Thermal TOFs (black line) over Pt(211) increases with $T$ but are vanishingly small at even the highest temperature. Oxygen atoms are the most abundant surface intermediate across this regime (Supplementary Fig. 5a), and by degree of rate control[34] analysis (Supplementary Fig. 6a), $N_2$ dissociative adsorption is rate-controlling, both consistent with the high $N_2$ dissociation barrier. The green region of Fig. 3b shows TOFs at $N_2$ and $O_2$ vibrational temperatures from 3000 to 10,000 K. TOFs are increased substantially relative to thermal-only catalysis, especially at the lowest bulk temperatures, i.e. gas and catalyst temperatures, and apparent activation energies are diminished. Predicted surface coverages are unchanged from the thermal case (Supplementary Fig. 5b) but the rate-controlling step changes to NO desorption at lower bulk and $O_2$ adsorption at higher bulk temperatures (Supplementary Fig. 6b), consistent with $N_2$ dissociation rates enhanced by vibrational excitation. The orange region of Fig. 3b shows TOFs at N radical density artificially set to 2 to 8 orders of magnitude less than the $N_2$ density and $O/O_2$ fraction to $10N/N_2$. N and O radicals have a similar to even greater enhancing impact on NO TOFs, most notably at the lowest bulk temperatures. NO is the most-abundant surface species at bulk temperatures below 800 K (Supplementary Fig. 5c), and NO desorption becomes rate controlling (Supplementary Fig. 6c), both reflecting the assumed barrierless accommodation of O and N by the Pt surface. At higher bulk temperatures, surface coverages tend to zero and rates become controlled by oxygen adsorption. Both vibrational excitation and radical adsorption relax the rate limitations of $N_2$ dissociation and are particularly effective at lower bulk temperatures.

Rates derived from kinetic parameters appropriate to a Pt terrace lead to similar general observations (Supplementary Fig. 7). Absolute Pt terrace TOFs, however, are significantly less than Pt steps at all but the highest vibrational temperatures or N densities, consistent with the greater reaction barriers on the terrace.

*Plasma vs plasma-catalytic NO production.* The above microkinetic models predict absolute, per active site rates at given

conditions[35]. Experiments most directly provide access to product concentrations rather than reaction rates. To compare plasma-only to plasma-catalytic productivity, we develop well-mixed, isothermal integral reactor models appropriate to the plasma afterglow region and the Pt catalyst bed, respectively (Fig. 4a and b and see details in Supplementary Methods)[26]. Bulk temperatures of the gas and catalyst are assumed to be 873 K to correspond with experiments.

We describe noncatalytic NO oxidation in the afterglow (Fig. 4a) using the Zeldovich mechanism, consistent with previous experiments and simulations of nonthermal radio frequency, gliding arc, and microwave $N_2/O_2$ plasmas[10–12,15,16,28,36]:

$$N_2(g) + O(g) \leftrightarrow NO(g) + N(g) \quad (3a)$$

$$O_2(g) + N(g) \leftrightarrow NO(g) + O(g) \quad (3b)$$

Rate constants are from experimental measurements (Supplementary Table 3)[15,37,38]. To capture the influence of vibrational excitations on the rate of Reaction (3a), we reduce the reaction barrier by an amount commensurate with the degree of plasma-induced vibrational excitation (Fig. 2, middle)[16]. We take the reactor length to be consistent with the length of the heating mantle and flow rates consistent with experiment.

To describe NO concentrations generated by the plasma and porous Pt catalyst together (Fig. 4b), we treat this region as a sum of coupled contributions of a noncatalytic bulk-phase, described using the same Zeldovich parameters, and a surface-catalytic phase, described using the Pt step parameters. The relative contributions of these two phases is a function of the reactor volume and number of active sites (Supplementary equation (11)). The total volume is taken to be that of the porous Pt catalyst bed, and flow rate and free volume-to-active site ratios are taken to be consistent with the experimental setup.

Figure 4c reports the plasma-only, noncatalytic NO concentrations, plotted as NO concentration for ready comparison to experiment, as a function of assumed vibrational temperature and inlet atomic N density as descriptors because of their relevance in plasma $N_2$ oxidation[10–12,15,16,28,36,39]. Wide ranges of $T_{vib}$ and $P_N$ are included to represent different plasma characteristics. The partial pressures of other species are reported in Supplementary Fig. 8. NO concentrations increase monotonically with $T_{vib}$ and

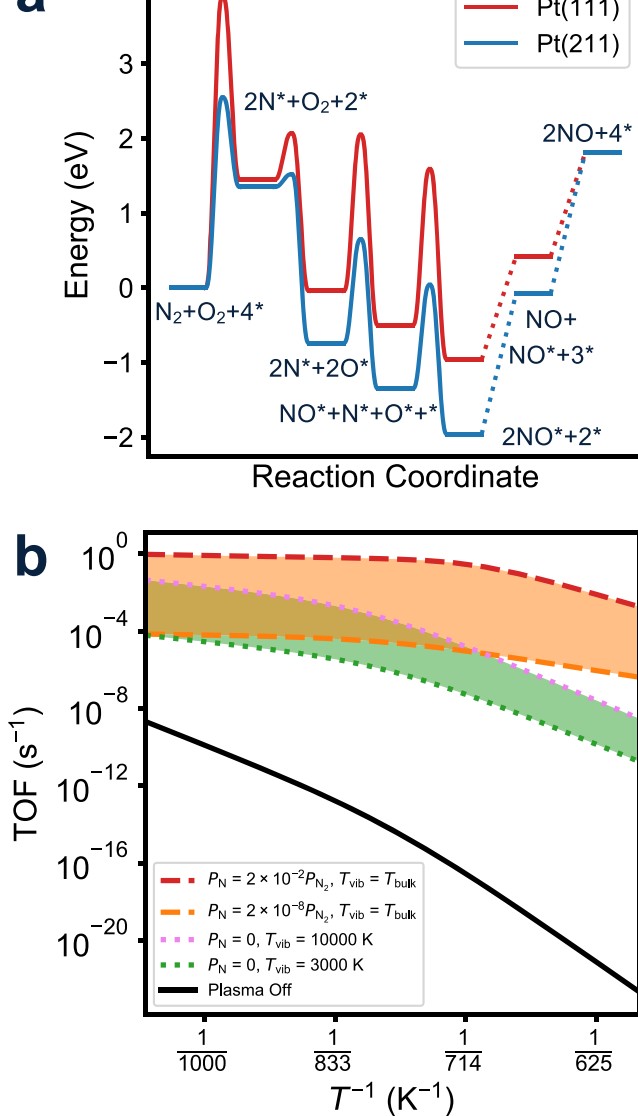

**Fig. 3 Microkinetic N₂ oxidation rates. a** DFT-reported potential energy surfaces for thermal N₂ oxidation on Pt(211) and Pt(111). **b** Reaction rates vs. reciprocal temperature on Pt(211) at various vibrational temperatures and N radical partial pressures. Pressures of N₂ and O₂ are 4.995 and 0.005 mbar, respectively.

concentrations greatly exceed plasma-only behavior at relatively high N radical densities and especially at the lower limit of vibrational temperatures, conditions typical of low-$P$ radio frequency and microwave plasmas[11,29,30].

The origins of this catalytic influence are revealed by an analysis of relative steady-state reaction fluxes normalized to the overall NO production rate[26,40]. Figure 5 reports these fluxes at four representative conditions corresponding to the four corners of Fig. 4d, focusing on pathways involving nitrogen. At a relatively low vibrational temperature and N radical density (Fig. 5c and lower left of Fig. 4d), the absolute TOF is only $5.3 \times 10^{-9} s^{-1}$ and is primarily limited by the rate of generation and adsorption of N radicals. At a much greater vibrational temperature and low N radical density (Fig. 5d and lower right of Fig. 4d), N₂ dissociative adsorption dominates surface accommodation of nitrogen and the absolute TOF increases to $4.5 \times 10^{-3} s^{-1}$. Both of these regimes are blue in Fig. 4e because the Zeldovich reaction in the larger homogeneous volume is more effective at producing NO than in the smaller catalyzed volume. At high vibrational temperature and N radical density (Fig. 5b and upper right corner of Fig. 4d), the absolute catalyzed TOF reaches its maximum of $0.22 \, s^{-1}$. Catalyzed rates are dominated by the large fluxes of N radicals to the catalyst and their subsequent reactions to NO or recombination to N₂. These surface reactions become more effective at directing N into channels that produce NO than is the homogeneous phase, where the reverse Zeldovich reaction depletes NO. As a result, this regime is yellow in Fig. 4e. At high N radical density but lower vibrational temperature (Fig. 5a and upper left corner of Fig. 4d), surface reactions remain dominated by N adsorption and reaction to NO or N₂. Despite the fact that the TOF in this quadrant $(0.13 \, s^{-1})$ is less than in the upper right, the plasma-catalyst combination is most effective in yielding NO, because the catalyst is most effective here in shunting N radicals away from the unproductive reverse Zeldovich reaction.

NO concentrations are maximized at high $T_{vib}$ and high N radical densities in the plasma-only and plasma-catalytic reactors, respectively (Fig. 4). In both of these regimes, a large fraction of plasma-generated N radicals ultimately return to N₂ either homogeneously or at the catalyst surface, diminishing the energy efficiency of NO production. To compare the theoretical energy consumption of the two reactors as a function of reaction species, we calculate the enthalpy required to reach $T_{vib}$ and $P_N$ from reactants at the bulk gas temperature, and normalize the energy to the NO production (details in Supplementary Methods). Supplementary Fig. 11 shows the plasma-only reactor is the most energy-efficient at higher $T_{vib}$, consistent with previous reports[16]. The catalyzed plasma reactor is more energy-efficient overall and is particularly efficient at low $T_{vib}$ and intermediate $P_N$, where the energy deposited into dissociated N₂ is most effectively directed into NO. The predicted energy consumption is two orders of magnitude lower than plasma only in this regime. The minimum predicted energy consumption is 2.9 MJ/mol$_{NO}$ at $T_{vib} = 1000$ K and $\frac{P_N}{P_{N_2}} = 10^{-4}$, which exceeds the minimum energy (about 0.3 MJ/mol$_{NO}$) using established estimates of the efficiencies of known processes in a N₂-O₂ NTP[16,41,42]. These results illustrate that the quest for optimal NO productivity and optimal energy efficiency may lead to different target plasma regimes.

**Optimal plasma-catalytic N₂ oxidation regimes.** Thermal N₂ oxidation rates and thus NO production are negligibly small at the conditions of Fig. 4. The results of Fig. 4e, then, highlight the existence of regimes in which plasma-catalytic N₂ oxidation is expected to be much more effective than plasma-only or thermal catalytic N₂ oxidation, qualitatively consistent with the results of

nonmonotonically with N radical density. To understand this behavior, we plot NO concentrations and the rate of each Zeldovich reaction as a function of $P_N$ in Supplementary Fig. 10. The net rates of $O_2 + N \leftrightarrow NO + O$ and $N_2 + O \leftrightarrow NO + N$ increase and decrease with increasing N radical densities, respectively. Because the rate of the second reaction is more sensitive than the first, overall NO production decreases with increasing $P_N$.

Figure 4d reports corresponding NO concentrations in the plasma-catalytic regime as a function of the same parameters. Complete results are summarized in Supplementary Fig. 8. NO concentrations are of similar orders of magnitude overall but less sensitive to vibrational temperature and more sensitive to N radical density. These differences become more evident when plotted as a ratio of concentrations, as shown in Fig. 4e. The plasma-catalyst combination is expected to have a negligible to deleterious effect on NO production across a large swath of N radical densities and vibrational temperatures. In contrast, NO

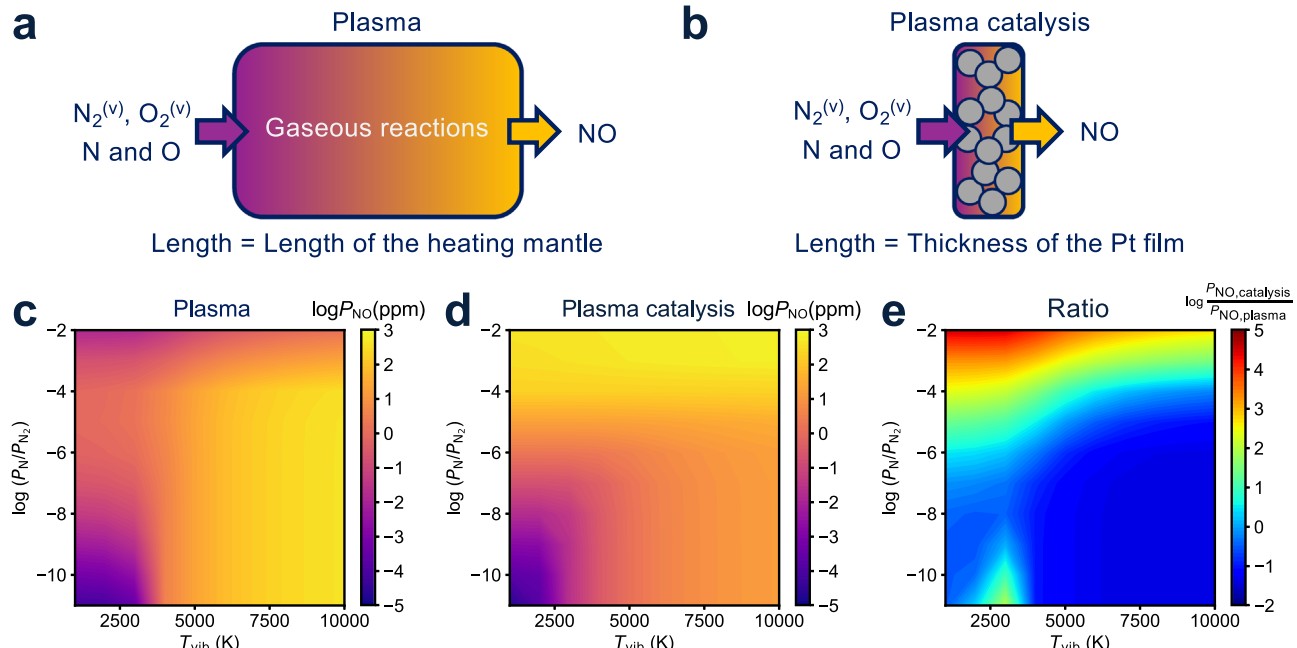

**Fig. 4 Plasma vs plasma-catalytic NO production.** Schematic representations of integral reactor models for (**a**) plasma reactions and (**b**) plasma catalytic reactions, respectively. Outlet NO concentrations (ppm) of (**c**) plasma reactions and (**d**) plasma catalytic reactions against $T_{vib}$ and $P_N$ at 873 K, where the inlet pressures of $N_2$ and $O_2$ are 4.995 and 0.005 mbar, respectively. The residence times are 0.28 s and 6.4 μs in (**a**) and (**b**), respectively. The number of active Pt sites is 230 nmol. **e** The ratio of NO produced in (**d**) and (**c**).

Fig. 1d. We next look to understand the observed sensitivity to $O_2$ fraction. Supplementary Fig. 12a reports the computed NO concentrations from a Pt catalyst modeled within the plasma catalytic reactor of Fig. 4b as a function of $O_2$ fraction. The predicted NO concentration varies non-monotonically across the experimental range of $O_2$ mole fractions, consistent with Fig. 1d. However, the predicted NO concentration maximizes at $O_2$ mole fraction $= 2 \times 10^{-3}$, greater than the experimental observation. Moreover, NO production decreases only minimally with $O_2$ mole fraction after reaching the maximum. Consistent with this model, catalytic rates at fixed reactant pressures (Supplementary Fig. 12c) are predicted to maximize at an $O_2$ mole fraction of $2 \times 10^{-2}$ and to decrease at higher mole fractions.

In the plasma-catalytic experiment of Fig. 1, the thin Pt film was placed at the center of the heating mantle. The plasma-generated species can react homogeneously in the space ahead of the catalyst bed, over the bed itself, and potentially even postcatalyst. To better represent the experimental configuration and to rationalize the dependence of integral NO production on the gas composition, we construct a series of precatalyst afterglow reactor, catalytic reactor, and postcatalyst reactor (Fig. 6a). The first reactor models the precatalyst region within the heating mantle. We explored a variety of radical densities and $T_{vib}$ fed to this first reactor based on reported variations in properties of $N_2$-$O_2$ plasmas with bulk gas composition. We predict NO concentrations across a wider gas composition range than the experiments in Fig. 1. The base properties, $P_N = 2 \times 10^{-3} P_{N_2}$ and $T_{vib} = 6000$ K, are based on observations of a $N_2$ radio frequency plasma operated at conditions similar to Fig. 1[29,30]. With increasing $O_2$ mole fractions, the vibrational temperature was reported to decrease, $\frac{P_N}{P_{N_2}}$ observed, and predicted to vary non-monotonically or decrease monotonically, and $\frac{P_O}{P_{O_2}}$ to remain almost constant[11,43–45]. We compare NO concentrations under various assumed sensitivities of vibrational temperature and radical densities to $O_2$ mole

fractions and report results in Supplementary Methods. The relationship between NO productivity and gas composition is weakly sensitive to these variations and tracks the impact on the plasma-only NO productivity; the largest impacts are on the composition at maximum NO production, which varies from $P_{O_2}/(P_{O_2} + P_{N_2}) = 10^{-3.3}$ to $10^{-2.9}$, and total NO production at the maximum, which varies from $10^{2.3}$ to $10^3$ ppm. Given this weak dependence, we focus the narrative on the simplest model, shown in Fig. 6; conclusions are insensitive to this choice.

Figure 6b compares the NO concentration vs $O_2$ fraction between the reactor series of Fig. 6a (red line) and the plasma-only model of Fig. 4a (blue line), plotted on a log scale. NO concentrations from both reactors exceed thermal $N_2$-$O_2$ equilibrium and generally increase with $O_2$ content. The reactor series captures the non-monotonic increase in NO concentration and better predicts the peak in concentration than does the single reactor model of Fig. 4b. The Pt catalyst enhances NO production substantially at $O_2$ fractions less than $3 \times 10^{-3}$. NO concentrations are sensitive to catalyst site number, but this effect saturates at about $10 n_{Pt}$, above which NO production varies minimally. At greater $O_2$ mole fractions, however, NO concentrations converge to those of the uncatalyzed reactor. To validate this prediction, we performed experiments at high $O_2$ pressures, combined with those of Fig. 1, and plotted on a logarithmic scale in Fig. 6c. The experimental observations are consistent with the series reactor predictions across a wide range of $N_2$-$O_2$ compositions, including the evident advantages of the catalyzed reactor, sensitivity to composition, and convergence to the plasma-only productivity at highest $O_2$ mole fractions. The series reactor model successfully captures the obvious drop of NO concentration after reaching the maximum at intermediate $O_2$ mole fractions and the increasing NO concentration at high $O_2$ mole fractions, which are not observed in the plasma catalytic model alone (Supplementary Fig. 12a). Models and experiments appear to differ in predicted plasma-only NO concentration sensitivity to $O_2$ mole fraction in

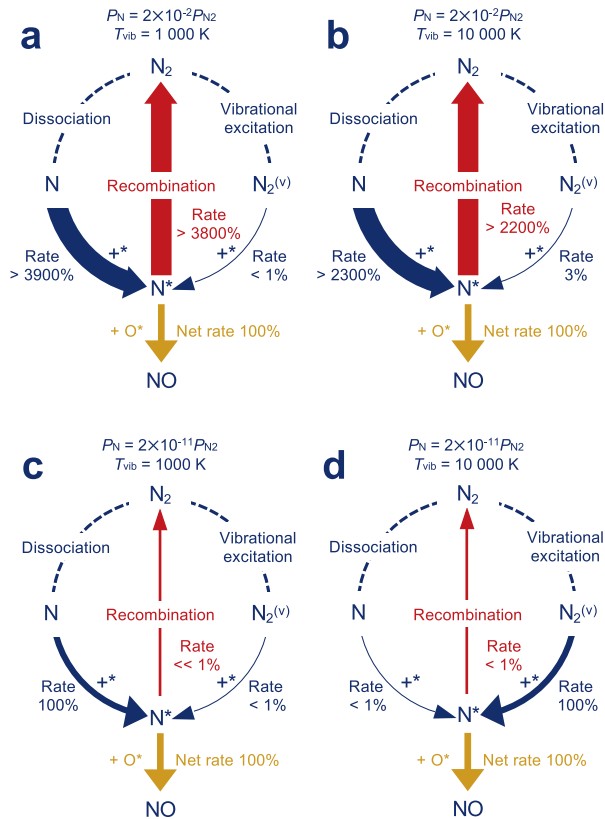

**Fig. 5 Reaction fluxes vs plasma conditions.** Steady-state reaction flux of $N_2$ oxidation on Pt(211) in a integral reactor at 873 K with (**a**) $P_N = 2\times10^{-2}P_{N_2}$ and $T_{vib} = 1000$ K, (**b**) $P_N = 2\times10^{-2}P_{N_2}$ and $T_{vib} = 10000$ K, (**c**) $P_N = 2\times10^{-11}P_{N_2}$ and $T_{vib} = 1000$ K, and (**d**) $P_N = 2\times10^{-11}P_{N_2}$ and $T_{vib} = 10000$ K, presented as Sankey diagrams, in which each segment indicates a reaction and the corresponding segment width qualitatively indicates the value of the rate normalized to the NO production rate. Solid lines represent reactions involving catalytic sites and dotted lines represent plasma processes. Blue, gold and red mean intermediate, productive and unproductive routes, respectively. The reaction conditions are consistent with the four corners of Fig. 4d.

the dilute limit, likely the result of some combination of experimental uncertainty in NO concentration at these very low levels and uncertainties in the reactor model details. The prominent role of catalyst in promoting NO production is robust to these details.

This loss in catalyst effectiveness can be understood from Supplementary Fig. 12b and d. Catalyzed NO production is maximized at conditions that balance the coverage of adsorbed N and O. At sufficiently high $O_2$ concentrations, the catalyst surface becomes O-covered and NO productivity decreases. That the same phenomenon is at play in the series reactor model is shown in Supplementary Fig. 13. As a result, the catalyst is less effective in promoting (or inhibiting) NO production, as observed in the experiments. X-ray photoelectron spectra of the Pt catalyst following exposure to plasmas from 1 to 15% $O_2$ overlap with fresh metallic Pt (Supplementary Fig. 3b), suggesting Pt remains metallic in a wide range of $O_2$ mole fractions. Deconvolution of the X-ray photoelectron spectrum of the Pt catalyst following exposure to the 20% $O_2$ plasma (Supplementary Fig. 3c) reveals features attributable to metallic Pt (13%) and oxidized

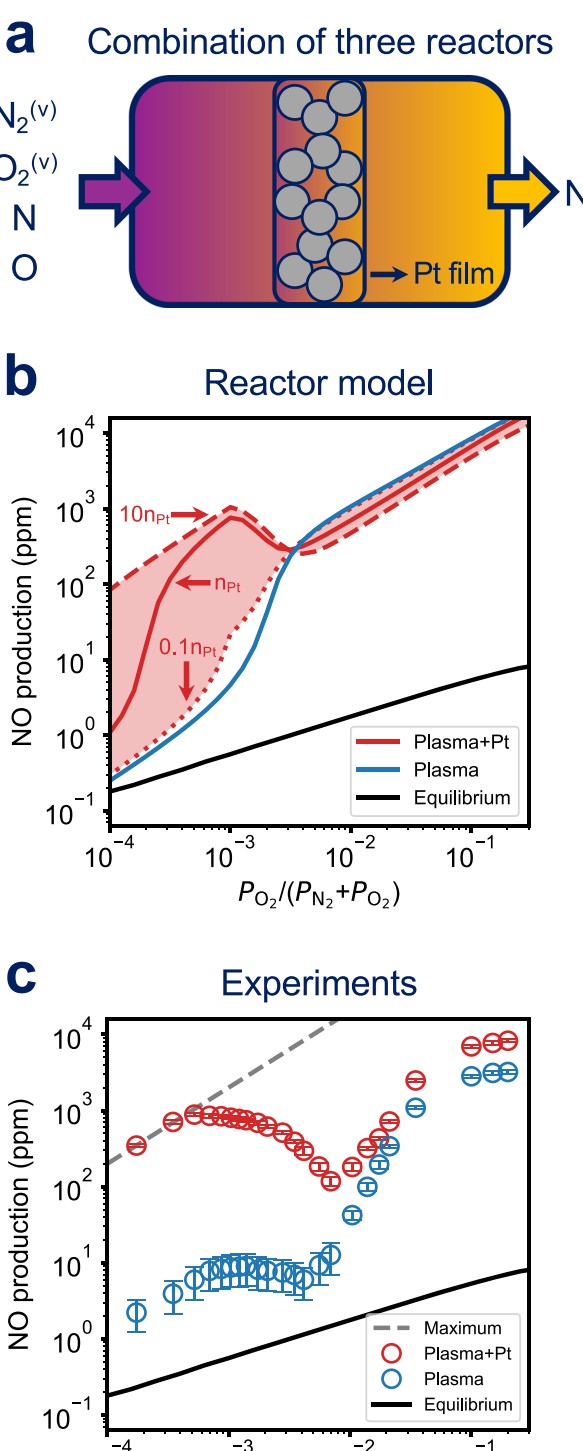

**Fig. 6 Composition sensitivity. a** Series of three integral reactors to describe the reactor of Fig. 1. **b** NO production as a function of $O_2$ pressure in plasma-only and plasma-catalytic reactor series at $P_N = 2\times10^{-3}P_{N_2}$, $T_{vib} = 6000$ K, and bulk temperature 873 K. Red solid line represents a Pt active site number of 230 nmol and dashed and dotted line represent order-of-magnitude variations in that number. **c** Observed NO production vs $O_2$ pressure ratio. Maximum represents all $O_2$ coverts to NO. Total reactor pressure is 5 mbar. The error bars represent the standard deviation of at least three measurements.

Pt (87% PtO and $PtO_2$). Nonetheless, NO production is observed to be increased by a factor of 2.6 over the background at $P_{O_2}/(P_{O_2} + P_{N_2}) > 0.20$ (Supplementary Fig. 15), comparable to a previous report[18], and suggesting that Pt oxides may contribute to NO production at higher $O_2$ exposures. The highest $N_2$ conversions to NO are 0.52% and 0.20% for coupled plasma and Pt and plasma only, respectively. Various plasma types show highest $N_2$ conversions from 0.1 to 14%[9]. These results illustrate how optimal catalyst characteristics are a function of reaction conditions.

Conventional heterogeneous catalyst design typically focuses on catalytic rates, as those (in concert with reaction conditions) tend to determine both productivity and energy efficiency. Coupling of NTPs to heterogeneous catalysts opens the potential to increase intrinsic rates[33] and, as illustrated here and elsewhere[46], to achieve product concentrations that exceed the limits of bulk thermodynamic equilibrium. Quantification of these effects has been challenged by the background influence of plasma- and/or catalyst-only reactions. We illustrate here an example of an NTP-catalyst combination that yields products at conditions at which neither alone is appreciably effective, providing unambiguous evidence for the benefits of the combination in the context of nitrogen fixation to NO.

The reactor model construction here illustrates the additional degrees of freedom that enter into optimizing an NTP-catalyst combination. Under conditions at which relevant reactions occur in the homogeneous and surface-catalyzed phases, optimal product concentrations do not necessarily correspond with optimal surface-catalyzed rates, and yield optimization involves tuning of catalyst properties, tuning of plasma-generated species, and tuning of the coupling of the two. Here that coupling is a function of the distance and volume of the pre-catalyst plasma, the volume of the catalyst bed, as well as active catalyst site density, which become additional design variables. For example, high active catalyst site density with a minimized catalytic volume lowers the impacts of gas-phase reactions and enables the exploration of intrinsic catalytic activities. The limitations of Pt suggest that catalysts that maintain low O and high N coverages at high $O_2$ mole fractions will be more effective for NO production in this operating regime. Coupling DFT computed kinetic parameters with the microkinetic strategy demonstrated here helps guide selections of catalysts. The strategy could be extended to systems where catalysts are placed in the plasma by including relevant reactions generating active plasma species[46,47]. Further, as shown here, yield and energy-efficiency do not necessarily correspond to the same regime of operation. The highest energy efficiency is predicted at intermediate $P_N$ and low $T_{vib}$. Tuning plasma properties or optimizing reactor configuration to control the densities of active species could lower energy consumption. These additional degrees of freedom both reflect the challenges and opportunities in selecting NTP and catalyst combinations optimally suited for target applications, especially at the small reactor scales most suited to NTPs. These results highlight the importance of constructing models that balance complexity and generality to guide exploration and optimization outside the regime of observations.

## Methods

The schematic of our plasma reactor is shown in Fig. 1 (in two modes of operation). The setup consists of an inductive coil, which is connected to the matching network of a radio frequency generator (13.56 MHz, 300 W maximum power rating Huttinger PFG 300 RF). The coil encloses a quartz tube of 40 mm outer diameter and 700 mm length, mounted on both ends to two vacuum flanges which also serve as a mechanical support for the quartz tube. The reactor is equipped with one inlet (for $O_2$ and $N_2$) and one outlet connected with Hiden Analytical Quadrupole Mass Spectrometer HAL 201RC. The detection limit of NO is 50 ppb. The temperature of the center part of the reactor is controlled by a heating mantle.

The catalyst (porous Pt film) is prepared on yttria-stabilized zirconia (YSZ) tube (Ortech, 2 mm thickness, 25 mm diameter, and 245 mm length) by brush painting the organometallic Pt paste (Fuel Cell Materials) followed by a heat treatment at 900 °C for 2 h in air. The crystal structure and purity of the catalyst were determined by X-ray diffraction (XRD, Bruker, Cu Kα radiation, λ = 1.54056 Å) in the Bragg-Brentano configuration. Diffractograms were collected at a scan rate of 0.02° in the 2θ range of 20°–90°. The surface composition of the Pt film was investigated by X-ray photoelectron spectroscopy (XPS, ThermoFisher Scientific, K-alpha instrument). The surface morphology of the as prepared catalyst (Pt/YSZ) was characterized using a scanning electron microscope (FEI Quanta 3D FEG instrument) at an acceleration voltage of 3-5 keV (Fig. 1b and c). The Pt film has a porous structure and consists a network of percolated particles of the order of micron and thickness is around $14\,\mu m$. SEM micrographs before and after plasma experiments show no difference since the Pt catalyst is 15 cm far from the tail of the active plasma area. This is in good agreement with the minimal temperature increase on the catalyst (i.e. 1–2 °C) upon plasma ignition. The visible tail of the active plasma area decreased with $O_2$ mole fractions and remained away from the catalyst (Supplementary Fig. 2). We estimated the time for gas-phase species to travel from the tail of the coil to the heating mantle to be 0.2 s. N and O densities in similar low-pressure $N_2$-$O_2$ discharge afterglows have previously been found to persist at least 1 s[11]. Vibrational temperatures, in contrast, are observed to decrease over this timescale in a low-pressure $N_2$ afterglow[48,49]. We estimated vibrational temperatures to be 10000 K in the radio frequency plasma[29] and select $T_{vib} = 6000$ K as representative of the temperature drop expected during flow to the catalyst. The catalyst geometrical area is 20 $cm^2$, while the loading is 5 mg of Pt per $cm^2$. The surface area of the catalyst was determined by hydrogen adsorption with potential deposition method[50] and it was found to be 230 nmol of Pt adsorption sites.

The experiments were performed by co-injecting and co-activating 5 mbar $N_2$ and $O_2$ (100 standard cubic centimeters per minute) by RF plasma source with plasma power as 80 W while maintaining 0 W reflected power through a tunable matching network. The calibrations to quantify the NO production and $O_2$ consumption were carried out by using 100 and 1000 ppm NO in He and 1% $O_2$ in He cylinders, respectively. In each case, the standard gas mixture was used without dilution and with He dilution in the levels of 25% and 50% keeping the flow rate constant. In all the cases, a linear relation between the signal level and amount of the gas in study, has been observed. The concentration of NO produced during plasma experiments is in good agreement (5%) with the oxygen level decrease. Experiments were repeated three times. $N_2O$ and $NO_2$ concentrations are negligibly small in both plasma-only and plasma-catalytic experiments across $O_2$ fraction from $10^{-4}$ to 0.2 (Supplementary Fig. 15b).

The reaction and activation energies of nitrogen oxidation were collected from literature, where the calculations were performed using DACAPO with core electrons described by Vanderbilt ultrasoft pseudopotentials and exchange and correlation effects described by the RPBE functional[24]. The standard entropies of gas molecules are from NIST-JANAF thermochemical tables[51] and entropies of adsorbates on Pt were estimated with harmonic oscillator model[52]. The rate constants of adsorption, surface and desorption reactions were estimated with transition state theory[35]. The steady-state surface coverages and rates were solved with a mean-field microkinetic model, as detailed in Supplementary Methods[53]. The steady-state surface coverages described by ordinary differential equations(ODEs) were firstly solved using a method that automatically switches between nonstiff (Adams) and stiff (BDF) solvers, as implemented in scipy.integrate.odeint in Python. The steady-state coverages were further converged with a system of algebraic equations using the coverages solved from ODE as initial guesses. Newton's method implemented in mpmath.findroot in Python was used.

To estimate the adsorption rate of vibrationally excited molecules, the dissociative adsorption rates were modified to be an explicit function of $N_2$ and $O_2$ vibrational states[33]. The non-Boltzmann population densities of vibrationally excited $N_2$ and $O_2$ states in the plasma were estimated using the Treanor formula at different vibrational temperatures[54,55]. The vibrational temperatures of $N_2$ and $O_2$ are assumed to be the same. The rate constant of each vibrational state is estimated individually and the overall rate constant is calculated with the summation of all vibrational states. The first ten vibrational excited states were included because of the depopulation of highly excited levels. See details in Supplementary Methods. The adsorption of N and O (Reaction (2a) and (2b)) were included in the plasma catalysis model. $\frac{P_O}{P_{O_2}}$ is set as $10\frac{P_N}{P_{N_2}}$ since $O_2$ dissociation fraction is observed to be about one order of magnitude higher than $N_2$ in $N_2$-$O_2$ plasmas[11,31,32,45].

Experimentally measured forward rate constants for reactions in the Zeldovich mechanism were used (Supplementary Table 3)[15]. The backward rate constants were calculated with standard free energies to enforce thermal consistency[51]. The calculated backward rate constants are in agreement with experimental measurements[37,38]. Vibrational excitation of $N_2$ and $O_2$ are included using the same methods in the plasma catalytic model. Integral reactor models to predict the NO concentrations of plasma reactions and coupled plasma and catalysts are detailed in Supplementary Methods. In these models, we assume the flow is well-mixed in the reactor. The ordinary equation of gas compositions and surface coverages were solved simultaneously. We also elaborate this modeling approach for materials selection in nitrogen oxidation[26].

## Data availability

The experimental and simulation data in this study have been deposited in the Zenodo database [https://doi.org/10.5281/zenodo.4624272][56]. The experimental data is included in both the Excel file and the Python scripts for figure creation.

## Code availability

The code in this study have been deposited in the Zenodo database [https://doi.org/10.5281/zenodo.4624272][56]. The repository includes Python scripts to generate the simulation data and the figures. Other utility functions to generate potential energy surfaces, catalytic turnover frequencies and product concentrations are also included.

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

## Acknowledgements

This project was supported by the U.S. Department of Energy, Office of Science, Office of Basic Energy Sciences, under Award DE-SC0021107 (W.F.S.), TKI-Energie from Toeslag voor Topconsortia voor Kennis en Innovatie (TKI) from the Ministry of Economic Affairs and Climate Policy (M.N.T.). The computing resources for this work were provided by the Notre Dame Center for Research Computing and National Energy Research Scientific Computing Center, a DOE Office of Science User Facility supported by the Office of Science of the U.S. Department of Energy under Contract No. DE-AC02-05CH11231 (W.F.S.). ISPT, University of Twente, Nouryon, OCI Nitrogen, Vopak and Yara are also acknowledged for their support in the project.

## Author contributions

All authors contributed to the conception of the research problem and approach. H.M. developed the microkinetic models. R.K.S. performed the experiments. H.M., R.K.S., S.W., M.C.M.vd.S., M.N.T. and W.F.S. co-wrote the manuscript.

## Competing interests

The authors declare no competing interests.
