## [Peer Review File · Nature Communications]

Title: Observation and rationalization of nitrogen oxidation enabled only by coupled plasma and catalystEditorial Note: This manuscript has been previously reviewed at another journal that is not operating a transparent peer review scheme. This document only contains reviewer comments and rebuttal letters for versions considered at *Nature Communications*.

REVIEWER COMMENTS

Reviewer #1 (Remarks to the Author):

The authors have addressed sufficiently all comments raised by this reviewer in the original review. The manuscript provides valuable insights on the origins of synergistic effects in plasma-catalysis and is hence suggested for publication in each current state.

Reviewer #2 (Remarks to the Author):

I appreciate the clarification and explanation from the authors. However, the major issues of this work are still there. I feel the significance of this work is quite limited to the field of plasma catalysis.

1. The XPS results show 13% of surface Pt was metallic and 87% was Pt oxides at 20% O₂ mole fraction. Looking at Figure S15, the NO production with Plasma+Pt is much higher than that with Plasma-only at a higher O₂ mole fraction. This result means that the surface Pt oxides could be more important for NO production. However, the model is only based on surface metallic Pt for a range of O₂ mole fractions and does not consider the oxidation of surface Pt sites when the O₂ mole fraction increases to >0.1. The reliability of this model for the prediction of NO production at a higher O₂ mole fraction is thus questionable. The robustness of the model is also limited.

2. The authors claim that the objective of this work is not to demonstrate a system that meets the goals of any particular practical application. The authors also highlight that the findings of this work can guide material selection and system optimization. However, I feel that these limited insights only apply to the low-pressure plasma system with a heated catalyst bed placed far from the plasma presented in this work. In my view, it is still not clear how we select an appropriate catalyst for this reaction. Should we select a metal catalyst or a metal oxide catalyst? If the catalyst is placed in the plasma, is the current suggestion still valid?

3. In the paper, the authors mention catalyst site density a few times, but do you mean Pt sites or Pt oxide sites?

4. In the abstract, the authors claim that Pt catalyzes NTP-generated radicals and vibrationally excited molecules to produce NO. Can vibrationally excited molecules reach the catalyst surface?

5. These results provide unambiguous evidence that productivity arises solely from the synergy between plasma and catalyst, an observation unique to the literature. Why this is considered unique to the

literature?

6. The authors claim that catalyst performance is sensitive to gas composition is an important observation. In my view, this is well known but this sensitivity has not been considered in this work (for example the model) when selecting a catalyst.

Reviewer #1:

The authors have addressed sufficiently all comments raised by this reviewer in the original review. The manuscript provides valuable insights on the origins of synergistic effects in plasma-catalysis and is hence suggested for publication in each current state.

Response:

We appreciate the reviewer's positive comments about our manuscript.

Reviewer #2:

I appreciate the clarification and explanation from the authors. However, the major issues of this work are still there. I feel the significance of this work is quite limited to the field of plasma catalysis.

Response:

We thank the reviewer for taking the time to evaluate our manuscript and for providing valuable comments.

Comment #1:

The XPS results show 13% of surface Pt was metallic and 87% was Pt oxides at 20% O₂ mole fraction. Looking at Figure S15, the NO production with Plasma+Pt is much higher than that with Plasma-only at a higher O₂ mole fraction. This result means that the surface Pt oxides could be more important for NO production. However, the model is only based on surface metallic Pt for a range of O₂ mole fractions and does not consider the oxidation of surface Pt sites when the O₂ mole fraction increases to >0.1. The reliability of this model for the prediction of NO production at a higher O₂ mole fraction is thus questionable. The robustness of the model is also limited.

Response:

To better probe the oxidation state of Pt at different O₂ mole fractions, we have performed XPS measurement of Pt after exposure of plasmas with 1, 5, 10, 15 and 20% O₂. Each sample was measured after plasma exposure of at least 30 min. The results are compared with freshly prepared Pt, as shown in Supplementary Figure 3. The spectra of different samples overlap, with the only exception of a 20% O₂ plasma. Pt remains metallic under exposure of plasmas with O₂ mole fractions as high as 15%. The current model based on metallic Pt is therefore reliable and robust in a wide range of O₂ mole fractions, i.e. 10⁻⁴-10^{-0.8}.

Addition, Page 20:

As a result, the catalyst is less effective in promoting (or inhibiting) NO production, as observed in the experiments. X-ray photoelectron spectra of the Pt catalyst following exposure to plasmas from 1 to 15% O₂ overlap with fresh metallic Pt (Supplementary Fig. 3b), suggesting Pt remains metallic in a wide range of O₂ mole fractions. Deconvolution of the X-ray photoelectron spectrum of the Pt catalyst following exposure to the 20% O₂ plasma (Supplementary Fig. 3c) reveals features attributable to metallic Pt (13%) and oxidized Pt (87% PtO and PtO₂).

Addition, Supplementary Figure 3, Supplementary Information:

Supplementary Figure 3: (a) X-ray diffraction of porous Pt film supported on yttria stabilized zirconia(YSZ). X-ray photoelectron spectroscopy (XPS) of (b) fresh Pt and Pt after exposure of 1, 5, 10 and 15% O₂ plasma and (c) Pt after exposure of 20% O₂ plasma for 3 h.

Comment #2:

The authors claim that the objective of this work is not to demonstrate a system that meets the goals of any particular practical application. The authors also highlight that the findings of this work can guide material selection and system optimization.

2.1 However, I feel that these limited insights only apply to the low-pressure plasma system with a heated catalyst bed placed far from the plasma presented in this work.

Response:

We have already addressed this point extensively in the previously revised Conclusions. Neither the type of plasma nor the exact plasma-catalyst configuration are explicit input parameters to the model presented here, which could be extended to other pressure regimes and configurations, and has indeed been extended to other materials (ref. 26). The particular experimental choices here are motivated by the fact that (i) low pressure nitrogen plasmas and their afterglows are well understood, so that densities and lifetime of various species are well known, (ii) the post-discharge (not the active plasma) is known for significant concentrations of vibrationally excited and atomic species, and (iii) the configuration reduces experimental complexity. We believe that demonstration of a robust and quantitative connection between experiment and microkinetic model within this regime is a significant step forward for the plasma catalysis field; indeed, it is a relatively rare achievement even within thermal catalysis.

2.2 In my view, it is still not clear how we select an appropriate catalyst for this reaction. Should we select a metal catalyst or a metal oxide catalyst?

Response:

We would first reemphasize that the goal of this work is not to identify an “appropriate” catalyst; no such claim is made at any point. Further, as is evident from the work here (and is in general true in all of heterogeneous catalysis) the appropriate catalyst is a function of the plasma properties and the desired objective. That dependence is only amplified when a catalyst and plasma are combined. This point is made, for instance, in contrasting the regimes of maximum catalytic efficiency (Figure 4) and maximum energy efficiency (Supplementary Figure 11). In our opinion, elucidation of the relationship between plasma characteristics and catalytic performance is of significant fundamental and practical import.

Further, as highlighted in the response above, all evidence points to metallic Pt as the active material in the chemistry reported here. Again as is true in heterogeneous catalysis in general, other classes of catalyst could be of interest. The work reported here is a foundation upon which to start to answer such questions.

2.3 If the catalyst is placed in the plasma, is the current suggestion still valid?

Response:

We are not entirely clear as to the meaning of “suggestion”. The results are reported and validated for the reactor configuration shown in Figure 1. We do not report experiments or models with the catalyst directly in the plasma. While we do not know, it is indeed possible that to achieve the level of agreement between experiment and model reported here for a configuration in which the catalyst and plasma are in direct contact would require some further elaboration of the model, for instance to include other plasma-generated species. With that said, prior models (e.g. ref. 33 and 46) have provided valuable guidance for configurations in which plasma and catalyst are in direct contact and based on model assumptions similar to those here.

Addition, Page 21:

Coupling DFT computed kinetic parameters with the microkinetic strategy demonstrated here helps guide selections of catalysts. The strategy could be extended to systems where catalysts are placed in the plasma by including relevant reactions generating active plasma species. [46,47]

[46] Mehta, P.; Barboun, P. M.; Engelmann, Y.; Go, D. B.; Bogaerts, A.; Schneider, W. F.; Hicks, J. C. Plasma-Catalytic Ammonia Synthesis beyond the Equilibrium Limit. *ACS Catal.* 2020, 10, 6726–6734.

[47] van ‘t Veer, K.; Engelmann, Y.; Reniers, F.; Bogaerts, A. Plasma-Catalytic Ammonia Synthesis in a DBD Plasma: Role of Microdischarges and Their Afterglows. *J. Phys. Chem. C* 2020, 124, 22871–22883.

Comment #3:

In the paper, the authors mention catalyst site density a few times, but do you mean Pt sites or Pt oxide sites?

Response:

We mention “site density” in the Conclusions section to illustrate that product concentration is a function of active catalyst site density, volume of the catalyst bed and other parameters. The site means active catalyst site in general and is not limited to Pt or Pt oxide here.

Modification, Page 21:

Here that coupling is a function of the distance and volume of the pre-catalyst space, the volume of the catalyst bed, as well as active catalyst site density, which become additional design variables. For example, high active catalyst site density with a minimized catalytic volume lowers the impacts of gas phase reactions and enables exploration of intrinsic catalytic activities.

Comment #4:

In the abstract, the authors claim that Pt catalyzes NTP-generated radicals and vibrationally excited molecules to produce NO. Can vibrationally excited molecules reach the catalyst surface?

Response:

Both experimental measurements and simulations suggest vibrational temperature decreases in low-pressure N₂ afterglows [48,49]. T_{vib} could drop a few thousands of degrees in 1 second but remains much higher than the bulk temperature. We estimated the time for gas-phase species to travel from the tail of the coil to the heating mantle to be 0.2 s. T_{vib} of radio frequency plasmas at similar conditions were observed to be higher than 10000 K [29]. The model-assumed $T_{\text{vib}} = 6000$ K accounts for this T_{vib} decrease during travel to the catalyst. We have discussed this question on Page 22 in the previous submission.

[29] Ricard, A., Sarrette, J. P., Oh, S. G., & Kim, Y. K. (2016). Comparison of the Active Species in the RF and Microwave Flowing Discharges of N₂ and Ar–20% N₂. *Plasma Chemistry and Plasma Processing*, 36(6), 1559-1570.

[48] Zhang, Q. Y., Shi, D. Q., Xu, W., Miao, C. Y., Ma, C. Y., Ren, C. S., ... & Yi, Z. (2015). Determination of vibrational and rotational temperatures in highly constricted nitrogen plasmas by fitting the second positive system of N₂ molecules. *AIP Advances*, 5(5), 057158.

[49] Loureiro, J., Sá, P. A., & Guerra, V. (2001). Role of long-lived N₂ (X¹Σ^{g+}, v) molecules and N₂ (A³Σ^{u+}) and N₂ (a'¹Σ^{u-}) states in the light emissions of an N₂ afterglow. *Journal of Physics D: Applied Physics*, 34(12), 1769.

Comment #5:

These results provide unambiguous evidence that productivity arises solely from the synergy between plasma and catalyst, an observation unique to the literature. Why this is considered unique to the literature?

Response:

In N₂ oxidation, NO yields have been reported to increase when plasmas are combined with a catalyst. In all cases that we are aware of, the observed improvements are with respect to significant plasma-only NO yield, as discussed on Page 4 of our previous submission. In many other reactions [1,2], plasma coupled with catalysts are often explored at conditions at which NTP or thermal catalytic productivity are non-negligible. Productivity arising solely from synergy between coupled plasma and catalyst is unique.

[1] Neyts, E. C.; Ostrikov, K. K.; Sunkara, M. K.; Bogaerts, A. Plasma Catalysis: Synergistic Effects at the Nanoscale. *Chem. Rev.* 2015, 115, 13408–13446.

[2] Whitehead, J. C. Plasma–Catalysis: the Known Knowns, the Known Unknowns and the Unknown Unknowns. *J. Phys. D: Appl. Phys.* 2016, 49, 243001.

Comment #6:

The authors claim that catalyst performance is sensitive to gas composition is an important observation. In my view, this is well known but this sensitivity has not been considered in this work (for example the model) when selecting a catalyst.

Response:

While sensitivity of catalyst performance to O₂-N₂ ratio has been explored previously [17,19], it has not been probed with the level of fidelity and in the low O₂ mole fraction regime we work in here. Further, one should draw a distinction between the external gas conditions and the active species in the plasma, which has not been probed previously. Again, “selecting a catalyst” is neither the point of the work nor a question that can be answered independent of the particulars of a plasma.

[17] Gicquel, A.; Cavadias, S.; Amouroux, J. Heterogeneous catalysis in low-pressure plasmas. *J. Phys. D: Appl. Phys.* 1986, 19, 2013–2042.

[19] Patil, B.; Cherkasov, N.; Lang, J.; Ibhaddon, A.; Hessel, V.; Wang, Q. Low temperature plasma-catalytic NO_x synthesis in a packed DBD reactor: Effect of support materials and supported active metal oxides. *Appl. Catal., B* 2016, 194, 123 – 133.

Manuscript Formatting Request

In addition to the above, you must comply with the following editorial requests; we will not be able to proceed with your revised manuscript otherwise. Please also see the Nature Communications formatting instructions, which you may find useful while preparing your revised manuscript.

Response:

We thank the editors for checking and providing valuable comments on the format of the manuscript. We have completed the editorial policy checklist, put raw experimental data in an Excel file and made it clear in the Data Availability, and explained all error bars in Figures.

Addition:

Page 25: The same Zenodo repository (<https://doi.org/10.5281/zenodo.4624272>) hosts the experimental and simulation data. The experimental data is included in both the Excel file and the Python scripts for figure creation.

Fig. 6 on Page 18: Total reactor pressure is 5 mbar. The error bars represent the standard deviation of at least three measurements.